# Neurodevelopmental Outcomes at 18 Months of Corrected Age for Late Preterm Infants Born at 34 and 35 Gestational Weeks

**DOI:** 10.3390/ijerph18020640

**Published:** 2021-01-13

**Authors:** Ruka Nakasone, Kazumichi Fujioka, Yuki Kyono, Asumi Yoshida, Takumi Kido, Shutaro Suga, Shinya Abe, Mariko Ashina, Kosuke Nishida, Kenji Tanimura, Hideto Yamada, Kandai Nozu, Kazumoto Iijima

**Affiliations:** 1Department of Pediatrics, Kobe University Graduate School of Medicine, Kobe 6500017, Japan; nakasone@med.kobe-u.ac.jp (R.N.); ykyono@med.kobe-u.ac.jp (Y.K.); yasumi@med.kobe-u.ac.jp (A.Y.); tkido@med.kobe-u.ac.jp (T.K.); sugashu@med.kobe-u.ac.jp (S.S.); sabe@med.kobe-u.ac.jp (S.A.); marikoa@med.kobe-u.ac.jp (M.A.); nk1125@med.kobe-u.ac.jp (K.N.); nozu@med.kobe-u.ac.jp (K.N.); iijima@med.kobe-u.ac.jp (K.I.); 2Department of Obstetrics and Gynecology, Kobe University Graduate School of Medicine, Kobe 6500017, Japan; taniken@med.kobe-u.ac.jp (K.T.); yhideto@med.kobe-u.ac.jp (H.Y.)

**Keywords:** late preterm infants, neurodevelopmental outcome, neurodevelopmental impairment

## Abstract

To date, the difference in neurodevelopmental outcomes between late preterm infants (LPI) born at 34 and 35 gestational weeks (LPI-34 and LPI-35, respectively) has not been elucidated. This retrospective study aimed to evaluate neurodevelopmental outcomes at 18 months of corrected age for LPI-34 and LPI-35, and to elucidate factors predicting neurodevelopmental impairment (NDI). Records of all LPI-34 (*n* = 93) and LPI-35 (*n* = 121) admitted to our facility from 2013 to 2017 were reviewed. Patients with congenital or chromosomal anomalies, severe neonatal asphyxia, and without developmental quotient (DQ) data were excluded. Psychomotor development was assessed as a DQ using the Kyoto Scale of Psychological Development at 18 months of corrected age. NDI was defined as DQ < 80 or when severe neurodevelopmental problems made neurodevelopmental assessment impossible. We compared the clinical characteristics and DQ values between LPI-34 (n = 62) and LPI-35 (*n* = 73). To elucidate the factors predicting NDI at 18 months of corrected age, we compared clinical factors between the NDI (*n* = 17) and non-NDI (*n* = 118) groups. No significant difference was observed in DQ values at 18 months of corrected age between the groups in each area and overall. Among clinical factors, male sex, intraventricular hemorrhage (IVH), hyperbilirubinemia, and severe hyperbilirubinemia had a higher prevalence in the NDI group than in the non-NDI group, and IVH and/or severe hyperbilirubinemia showed the highest Youden Index values for predicting NDI. Based on the results of this study, we can conclude that no significant difference in neurodevelopmental outcomes at 18 months of corrected age was observed between LPI-34 and LPI-35. Patients with severe hyperbilirubinemia and/or IVH should be considered to be at high risk for developing NDI.

## 1. Introduction

Despite the decrease in birth rate in recent years, the birth rate of late preterm infants (LPI) born between 34 0/7 and 36 6/7 gestational weeks (GW) is increasing [1,2]. LPI exhibit increased risks for early postnatal complications, such as respiratory disorders, feeding difficulty, hypoglycemia, jaundice, and hypothermia, compared with term infants [3]. We previously reported the effects of LPI birth on physical development in a longitudinal population-based study of 26,970 neonates born between 34 and 41 GW in Kobe City between 2006 and 2008. In that study, we classified subjects into LPI (*n* = 1414) and term infants (*n* = 25,556) and observed that the incidence of short stature at three years of age in the LPI group (2.9%) was significantly higher than that in the term group (1.4%) [4].

Regarding neurodevelopmental outcomes, the late prematurity period is described as a “critical period” for brain development. Brain weight at 34 GW is only 65% of the brain weight at term, and the cortical volume increases by 50% between 34 and 40 GW; thus, interruption of the in utero maturation process of the brain may contribute to neurodevelopmental impairment (NDI) [5]. To assess the risk of NDI, Petrini et al. conducted a retrospective cohort study in a Northern Californian population (141,321 children ≥ 30 GW born between 2000 and 2004) and observed that LPI had significantly higher risks of developing cerebral palsy and developmental delay/ intellectual disability than those born at term [6]. For school-age outcomes, Morse et al. reported that the risk for developmental delay or disability was 36% higher among LPI than term infants for the first five years of life [7]. Concerning adulthood outcomes, Moster et al. reported that LPI had an increased risk for neurodevelopmental problems, including cerebral palsy, intellectual disability, and disorders of psychological development, behavior, and emotion [8].

Thus, we have regarded late preterm births as a risk factor for the development of NDI. In our institute, we have routinely evaluated LPI born at 34 and 35 GW (LPI-34 and LPI-35, respectively) for their developmental quotient (DQ) at around 18 months of corrected age. However, a detailed neurodevelopmental evaluation for all these uncomplicated infants requires significant manpower and time, which is not available in resource-limited settings. Recently, Malgorzata et al. reported that LPI-34 and infants born at ≤33 GW should be classified into a more immature group [9]. Similarly, a cohort study in Australia studying the risk of mortality according gestational age classified infants born at 32 to 34 GW as moderate preterm, and those born at 35 to 36 GW as late preterm [10]; similarly, EPIPAGE-2, a cohort study of preterm infants in France, included infants born at 22 to 34 GW [11]. In addition, Kotecha et al. reported that preterm infants born at 33 to 34 GW had poorer lung function at 8–9 years of age than term infants, as did infants born at 25 to 32 GW [12]. Similarly, the risk of neonatal mortality in infants born at 35 GW has been reported to be 32% lower than that in infants born at 34 GW [13]. As mentioned above, several studies have treated LPI-34 separately from LPI-35 or older [9,10,11,12]. Therefore, we recalled the concept that routine evaluation of the psychomotor development of uncomplicated LPI could be limited only to those at high risk for NDI, such as infants born at 34 GW.

However, to date, many studies that examined long-term outcomes have categorized preterm infants born at 34–36 GW into the same LPI group, and even fewer studies have examined the prognosis of LPI according to gestational age. Thus, this study aimed to clarify the difference in neurodevelopmental outcomes at 18 months of corrected age between LPI born at 34 and 35 GW.

## 2. Patients and Methods

Gestational age was determined based on a dating ultrasound scan performed in the first trimester. Records of 214 LPI-34 (34 + 0 to 34 + 6, *n* = 93) or LPI-35 (35 + 0 to 35 + 6, *n* = 121) admitted to our facility between 2013 and 2017 were reviewed. Patients with congenital or chromosomal anomalies (total, *n* = 8; LPI-34, *n* = 3; LPI-35, *n* = 5), severe neonatal asphyxia (Apgar score ≤3 at 5 min, total, *n* = 3; LPI-34, *n* = 1; LPI-35, *n* = 2), and without DQ data (total, *n* = 68; LPI-34, *n* = 27; LPI-35, *n* = 41) were excluded (Figure 1). Psychomotor development was assessed as a DQ using the Kyoto Scale of Psychological Development (KSPD) at 18 months of corrected age (total, *n* = 135; LPI-34, *n* = 62; LPI-35, *n* = 73). The DQ was calculated by dividing the developmental age by the corrected age for prematurity and subsequently multiplying the quotient by 100 [14]. In this study, NDI was defined as a DQ < 80 or a condition wherein neurodevelopmental assessment was impossible due to severe central nervous system diseases, such as cerebral palsy, intellectual disability, global developmental delay, and/or non-verbal autism, at 18 months of corrected age [15].

Maternal, neonatal, placental histology, and psychomotor development data were collected from electronic medical records. Maternal data included maternal age, threatened premature labor (conditions causing subjective symptoms of uterine pain, contraction, bleeding, and/or shortening of uterine cervical length and, therefore, requiring tocolytic agents [16]), premature rupture of membrane (>24 h before delivery [16]), hypertensive disorder of pregnancy (maternal systolic blood pressure >140 mmHg and/or diastolic pressure >90 mmHg during pregnancy [17]), gestational diabetes (at least one value of the 75-g oral glucose tolerance test exceeding the threshold [18]), smoking (maternal smoking during pregnancy), multiple births, and birth by cesarean section. Neonatal data included sex, birthweight, Apgar scores at 1 and 5 min, small for gestational age infants (birthweight <10th percentile for gestational age [17]), artificial mechanical ventilation (regardless of the ventilation mode), respiratory distress syndrome diagnosed from chest X-ray findings, hyperbilirubinemia (requiring standard phototherapy based on our treatment criteria; spectral power: 10–15 µW/cm^2^/nm [19]), severe hyperbilirubinemia (requiring intensive phototherapy based on our treatment criteria; spectral power: 30–35 µW/cm^2^/nm [19]), hypoglycemia (initial serum glucose <40 mg/dL, requiring intravenous glucose infusion), intraventricular hemorrhage (IVH, all grades of IVH were included), and periventricular leukomalacia diagnosed by cranial ultrasonography. Placental histology data included histological chorioamnionitis (defined as neutrophil infiltration of amniotic membranes, umbilical cord, or chorionic plate [20]), funisitis (considered if neutrophil infiltration of umbilical vessel walls or Wharton’s jelly was observed [20]), infarction, and calcification. Psychomotor development data were recorded as DQs in three areas, postural-motor, cognitive-adaptive, and language-social, as well as overall DQ [21].

First, we compared clinical characteristics, including psychomotor development data, between the LPI-34 and LPI-35 groups. Second, we compared maternal, neonatal, and placental histology data between the NDI and non-NDI groups to screen for variables with *p*-values <0.05. In subsequent analysis, using these variables, we calculated the sensitivity, specificity, and Youden Index using receiver operating characteristic (ROC) curve analysis to determine the clinical predictors of NDI at 18 months of corrected age. The Youden index is the point farthest from the boundary delineating the area under the curve (0.500 on the ROC curve) and represents the (sensitivity + specificity − 1) value [15,22].

Data are expressed as median (range), mean ± standard deviation, or number (percent). To compare data from the LPI-34 and LPI-35 groups, the Mann–Whitney U test for maternal and neonatal numerical data, chi-squared test for maternal, neonatal, and placental categorical data, and Student’s *t*-test for psychomotor development data were used. Differences were considered to be statistically significant at *p* < 0.01. Analyses were performed using GraphPad Prism version 7.00 (GraphPad Software, La Jolla, CA, USA). Sample size calculations were performed using G*Power 3.1 (The G*Power Team, Heinrich-Heine-Universität Düsseldorf, Düsseldorf, Germany; http://www.gpower.hhu.de/) based on the hypothesis that the prevalence of infants with DQ <80 (NDI) in the LPI-35 group is approximately 10%, similar to that in term infants. If the prevalence of NDI was calculated to differ by 20% between the LPI-34 and LPI-35 groups, this would result in an 80% chance of detecting this difference with a type I risk error of 5% with 56 infants in each group.

## 3. Results

### 3.1. Patient Characteristics

There were 62 and 73 subjects in the LPI-34 and LPI-35 groups, respectively (Figure 1). When we compared the baseline characteristics of those with and without follow-up, no significant differences were observed between the LPI-34 and LPI-35 groups, respectively (Appendix A).

### 3.2. Clinical Characteristics in LPI-34 and LPI-35 Infants

Clinical characteristics of the study population are summarized in Table 1. The birthweight was significantly lower, whereas the incidence of cesarean section, artificial mechanical ventilation, respiratory distress syndrome, hyperbilirubinemia, and length of hospital stay were significantly higher in the LPI-34 group than in the LPI-35 group (all *p* < 0.01). Four infants in the LPI-34 group and one infant in the LPI-35 group could not be examined using the KSPD because of severe NDI (e.g., severe global developmental delay or non-verbal autism). No significant difference was observed between groups in DQ values at 18 months of corrected age for psychomotor development in the postural-motor, cognitive-adaptive, and language-social areas and overall DQ (Table 1). 

### 3.3. Clinical Characteristics in NDI and Non-NDI Infants

To elucidate the factors predicting NDI at 18 months of corrected age, we compared the clinical factors between the NDI (*n* = 17) and non-NDI (*n* = 118) groups (Table 2). Among the factors, IVH, male sex, hyperbilirubinemia, and severe hyperbilirubinemia had higher prevalence in the NDI group than in the non-NDI group (*p* < 0.05, Table 2). 

### 3.4. Sensitivity, Specificity, and Youden Index for Predicting NDI at 18 Months of Corrected Age

The sensitivity, specificity, and Youden Index that predicted NDI based on these four factors were examined (Table 3). IVH and/or severe hyperbilirubinemia showed the highest Youden Index values (0.284, Table 3).

## 4. Discussion

In this study, DQ values at 18 months of corrected age were similar between the LPI-34 and LPI-35 groups in each area and overall. IVH and/or severe hyperbilirubinemia were identified as clinical factors associated with a DQ < 80 in our cohort.

To date, some studies have compared LPI-34 and LPI-35 for early and late complications. Regarding early postnatal complications, Bastek et al. reported that acute complications such as IVH, seizure, apnea, respiratory assistance, necrotizing enterocolitis, reflux, hypoglycemia, feeding intolerance, sepsis, jaundice, anemia, and temperature instability were seven times higher in LPI-34 and three times higher in LPI-35 than those born after 39 GW (*p* < 0.001 and *p* < 0.05, respectively) [23]. In our study, artificial mechanical ventilation, respiratory distress syndrome, hyperbilirubinemia, and hypoglycemia were significantly higher in the LPI-34 group than in the LPI-35 group, which is consistent with the results of previous studies [23].

Moreover, a few investigations compared the long-term neurodevelopmental outcomes of LPI-34 and LPI-35. Morse et al. conducted a cohort study examining early school-age outcomes of LPI and reported that the proportion of infants with NDI gradually decreased with increasing gestational age. They revealed that the number of children who were judged as “not ready to start school” before entering elementary school was significantly higher in the LPI-34 group than in the LPI-35 and older groups [7]. In addition, Lipkind et al. reported that English and math test scores at school age showed significant increases with higher gestational age among infants who were born between 34 and 39 GW [24]. These previous reports have suggested worse neurodevelopmental outcomes in LPI-34 than in LPI-35.

In this study, we did not observe any difference in DQ values at 18 months of corrected age between the LPI-34 and LPI 35 groups in each area and overall. Similarly, Huddy et al. reported no significant difference in the incidence of behavioral problems during school life between LPI-34 and LPI-35, based on their study investigating the neurological prognosis at seven years of age in children born between 32 and 35 GW [25]. The present study excluded patients with congenital or chromosomal anomalies and severe neonatal asphyxia (Apgar score ≤3 at 5 min), which are known risk factors for NDI. Because infants with major anomalies tend to be born prematurely, and lower Apgar scores are more common in preterm births, these factors may have contributed to the increase in NDI in LPI with lower gestational age. Because our study excluded these “complicated” LPI, we could reliably compare the risk of NDI between “uncomplicated” LPI-34 and LPI-35.

In this study, severe hyperbilirubinemia and/or IVH were predictors of NDI. The incidence of IVH is reportedly higher in LPI than in term infants [26,27], but lower than in preterm infants born at <34 GW [23]. We found three grade I IVH cases in our cohort at a frequency (3/214, 1.4%) similar to those in previous reports [23]. Regarding the effect of IVH on neurodevelopmental prognosis, grade I–II IVH, even with no documented white matter injury or other late ultrasound abnormalities, is associated with adverse neurodevelopmental outcomes in extremely preterm infants [28]. Although no study has examined the effect of IVH on the neurological prognosis in LPI, even low-grade IVH may be associated with NDI in LPI. Although hyperbilirubinemia is a well-known complication among LPI [3,23], its neurological effect on LPI has not been adequately assessed. In a population-based study from Denmark, neonatal jaundice in children born at term was associated with psychological development disorders [29]. In a Canadian prospective cohort study targeting infants born at ≥35 GW, an association with attention-deficit disorder in the severe hyperbilirubinemia group and with autism in the combined moderate and severe hyperbilirubinemia group was reported, despite no incidence of kernicterus [30]. In our study, severe hyperbilirubinemia showed high specificity for predicting NDI, which may reflect the severity of bilirubin-induced neurological dysfunction [31]. 

Initially, we designed this retrospective study with the idea that excluding LPI-35 from a regular neurodevelopmental follow-up protocol would be clinically helpful, considering the limited manpower at follow-up. However, unexpectedly, LPI-35 were found to have an NDI risk similar to LPI-34, indicating that these subjects also need to be closely followed. Although not fully clarified in this study, it is necessary to elucidate all factors, including neonatal jaundice and IVH, that could predict NDI in these LPI-34 and LPI-35. As a result, it may be possible to make effective use of limited human resources by stratifying the follow-up targets and differentiating the degree of follow-up necessary.

This study had some limitations. First, the number of patients in this cohort was extremely limited for a population as large as the Japanese population due to the retrospective, single-center study design. Because the neurodevelopmental prognosis of uncomplicated LPI was generally favorable, there were high dropout rates during follow-up. Second, to clarify the necessity for routine neurodevelopmental assessment of “uncomplicated” LPI, we excluded infants with congenital or chromosomal anomalies and severe neonatal asphyxia, which are subject to careful follow-up regardless of gestational age. However, we believe that these exclusion criteria were necessary to reliably compare the risk of NDI between LPI-34 and LPI-35. Thus, we could not perform a multivariate analysis to determine the independent risk factors for NDI due to the small number of NDI patients. Third, although it has been suggested that placental pathology is involved in the development of NDI in preterm infants [32,33,34], we could not examine the placental histology of all enrolled infants in this study. Similar to a previous study by Ylijoki et al. [35], placental histology data were not associated with the risk of NDI in our population, despite the limited number of examined patients. However, because abnormal placental findings—especially vascular abnormalities—have been suggested to significantly affect abnormal neurodevelopment [36,37], the correlation between placental characteristics and NDI in LPI cannot be confirmed from this study alone. Thus, further prospective studies with large patient cohorts and detailed placental data, including LPI born at 36 GW and a control group of term infants, are required to determine the benefit of routine neurodevelopmental evaluation in detecting NDI for both LPI-34 and LPI-35, and to confirm the effect of severe hyperbilirubinemia and IVH on NDI in LPI.

## 5. Conclusions

No significant difference in neurodevelopmental outcomes at 18 months of corrected age was observed between LPI-34 and LPI-35 in our retrospective study. From a realistic perspective of daily obstetric practice, a difference of one week may not significantly affect neurodevelopmental outcome after 34 GW. Additionally, patients with severe hyperbilirubinemia and/or IVH may be considered at high risk of developing NDI.

## Figures and Tables

**Figure 1 ijerph-18-00640-f001:**
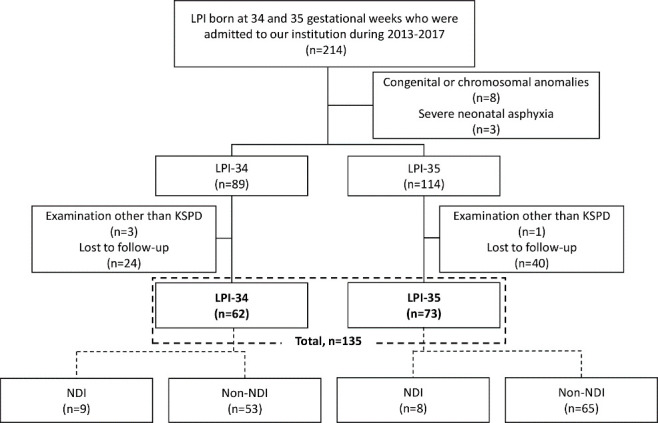
Flowchart of the patient selection process. In this study, neurodevelopmental impairment (NDI) was defined as developmental quotient (DQ) < 80 or a condition wherein neurodevelopmental assessment was impossible due to severe central nervous system diseases such as cerebral palsy, intellectual disability, global developmental delay, and non-verbal autism at 18 months of corrected age. KSPD, Kyoto Scale of Psychological Development; LPI, late preterm infant; LPI-34, LPI born at 34 gestational weeks; LPI-35, LPI born at 35 gestational weeks.

**Table 1 ijerph-18-00640-t001:** Clinical characteristics of the late preterm infants born at 34 and 35 gestational weeks.

	LPI-34 *n* = 62	LPI-35 *n* = 73	*p* Value
**Maternal Data**			
Maternal age, years	34 (18–42)	35 (18–41) *	0.22
Threatened preterm labor	36 (58)	44 (60)	0.79
Premature rupture of membrane	16 (26)	25 (35)	0.29
Hypertensive disorder of pregnancy	13 (21)	10 (14)	0.26
Gestational diabetes	9 (15)	15 (21)	0.36
Smoking	0 (0)	2 (3)	0.19
Multiple pregnancies	24 (39)	16 (22)	0.03
Cesarean section	53 (85)	45 (62)	<0.005
Outborn	3 (5)	7 (10)	0.29
**Neonatal Data**			
Male	39 (63)	42 (58)	0.53
BW, g	2061 (1354–2798)	2328 (1700–3562)	0.0001
Apgar score at 1 min	8 (3–10)	8 (2–9)	0.46
Apgar score at 5 min	9 (6–10)	9 (4–10)	0.44
Small for gestational age infants	14 (23)	10 (14)	0.18
Artificial mechanical ventilation	44 (71)	33 (45)	<0.005
Respiratory distress syndrome	17 (27)	6 (8)	<0.005
Hyperbilirubinemia	47 (78) #	40 (55)	<0.005
Severe hyperbilirubinemia	4 (7) #	8 (11)	0.39
Hypoglycemia	40 (67) #	34 (47)	0.02
Intraventricular hemorrhage	2 (3)	1 (1)	0.47
Periventricular leukomalacia	1 (2)	0 (0)	0.28
Length of hospital stay, days	29.5 (17–91)	19 (9–60)	<0.001
**Placental Characteristics**			
Chorioamnionitis ¥	10 (27)	9 (20)	0.93
Funisitis ¥	3 (8)	2 (4)	0.49
Infarction ¥	6 (16)	12 (27)	0.26
Calcification ¥	13 (35)	19 (42)	0.51
**Psychomotor Development Data**			
All areas $	92 ± 9	92 ± 11	0.74
Postural-motor $	89 ± 13	90 ± 13	0.71
Cognitive-adaptive $	93 ± 12	93 ± 13	>0.99
Language-social $	90 ± 11	88 ± 12	0.37
Neurodevelopmental impairment (DQ <80)	9 (15)	8 (11)	0.53

Data are expressed as median (range), mean ± standard deviation, or number (%). All intraventricular hemorrhage were at grade I. BW, birthweight; DQ, developmental quotient; LPI-34, late preterm infants born at 34 gestational weeks; LPI-35, late preterm infants born at 35 gestational weeks. * *n* = 72 (one missing data), # n = 60 (two missing data), $ *n* = 58 in LPI-34, and *n* = 72 in LPI-35 for DQ values due to data unavailability. ¥ *n* = 37 in LPI-34 and *n* = 45 in LPI-35 for placental characteristics due to inadequate histology data.

**Table 2 ijerph-18-00640-t002:** Clinical characteristics of the neurodevelopmental impairment (NDI) and non-NDI groups.

Characteristic	NDI *n* = 17	Non-NDI *n* = 118	*p* Value
**Maternal Data**			
Maternal age, years	35 (29–42)	35 (18–42) *	0.41
Threatened preterm labor	10 (59)	70 (59)	0.97
Premature rupture of membrane	3 (18)	38 (32)	0.22
Hypertensive disorder of pregnancy	3 (18)	21 (18)	0.99
Gestational diabetes	4 (24)	20 (17)	0.51
Smoking	0 (0)	2 (2)	0.59
Multiple pregnancies	2 (12)	38 (32)	0.08
Cesarean section	12 (71)	86 (73)	0.84
Outborn	1 (6)	9 (8)	0.80
**Neonatal Data**			
LPI-34	9 (53)	53 (45)	0.53
Male	14 (82)	67 (57)	0.04
Gestational age, weeks	34 (34–35)	35 (34–35)	0.61
BW, g	2224 (1512–3562)	2155 (1354–3540)	0.73
Apgar score at 1 min	8 (3–10)	8 (2–9)	0.21
Apgar score at 5 min	9 (6–10)	9 (4–10)	0.97
Small for gestational age infants	3 (18)	21 (18)	0.99
Artificial mechanical ventilation	12 (71)	65 (55)	0.23
Respiratory distress syndrome	3 (18)	20 (17)	0.94
Hyperbilirubinemia	15 (88)	72 (62) #	0.03
Severe hyperbilirubinemia	4 (24)	8 (7) #	0.03
Hypoglycemia	10 (59)	64 (55) #	0.78
Intraventricular hemorrhage	3 (18)	0 (0)	<0.001
Periventricular leukomalacia	0 (0)	1 (1)	0.70
Length of hospital stay, days	25 (10–91)	21.5 (9–65)	0.15
**Placental Characteristics**			
Chorioamnionitis ¥	2 (17)	17 (24)	0.56
Funisitis ¥	1 (8)	4 (6)	0.73
Infarction ¥	3 (25)	15 (21)	0.78
Calcification ¥	4 (33)	28 (40)	0.66
**Psychomotor Development Data**			
All areas	72 ± 5	94 ± 8	<0.001
Postural-motor	76 ± 14	90 ± 12	<0.001
Cognitive-adaptive	71 ± 4	95 ± 11	<0.001
Language-social	75 ± 10	90 ± 11	<0.001

Data are expressed as median (range), mean ± standard deviation, or number (%). All intraventricular hemorrhages were at grade I. BW, birthweight; LPI-34, late preterm infants born at 34 gestational weeks; NDI, neurodevelopmental impairment. * *n* = 117 (one missing data point), # *n* = 116 (two missing data points), $ *n* = 12 in the NDI group for developmental quotient values due to impossible psychomotor developmental assessment. ¥ *n* = 12 in the NDI and *n* = 70 in the non-NDI groups for placental characteristics due to inadequate histology data.

**Table 3 ijerph-18-00640-t003:** Sensitivity, specificity, and Youden index for predicting neurodevelopmental impairment at 18 months of corrected age.

	Sensitivity	Specificity	Youden Index	*p* Value
Male	0.824	0.432	0.256	0.04
Hyperbilirubinemia	0.882	0.379	0.261	0.03
Severe hyperbilirubinemia	0.235	0.931	0.166	0.03
IVH	0.177	1.000	0.177	<0.001
Male and/or IVH	0.824	0.432	0.256	0.04
Male and/or hyperbilirubinemia	0.941	0.155	0.096	0.29
Male and/or severe hyperbilirubinemia	0.882	0.397	0.279	0.03
Hyperbilirubinemia and/or severe hyperbilirubinemia	0.882	0.379	0.261	0.03
Hyperbilirubinemia and/or IVH	0.882	0.379	0.261	0.03
**Severe hyperbilirubinemia and/or IVH**	**0.353**	**0.931**	**0.284**	**<0.001**
Male and/or hyperbilirubinemia and/or severe hyperbilirubinemia	0.941	0.155	0.096	0.29
Male and/or hyperbilirubinemia and/or IVH	0.941	0.155	0.096	0.29
Male and/or severe hyperbilirubinemia and/or IVH	0.882	0.397	0.279	0.03
Hyperbilirubinemia and/or severe hyperbilirubinemia and/or IVH	0.882	0.379	0.261	0.03
Male and/or hyperbilirubinemia and/or severe hyperbilirubinemia and/or IVH	0.941	0.155	0.096	0.29

Emboldened texts are factors with the highest Youden Index. IVH, intraventricular hemorrhage.

## Data Availability

The data presented in this study are available in article.

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
