# Peer review of "Neurodevelopmental Outcomes at 18 Months of Corrected Age for Late Preterm Infants Born at 34 and 35 Gestational Weeks"

_ijerph, 2021, doi:10.3390/ijerph18020640_

Round 1
Reviewer 1 Report
The manuscript presents a retrospective analysis of the neurodevelopmental outcomes of late preterm infants. The study concludes that there is no significant difference between neurodevelopmental outcomes of 34 week infants as compared to those of 35 week infants when assessed at 18 months corrected age. Additionally, it concludes that severe hyperbilirubinemia and intraventricular hemorrhage are risk factors for neurodevelopmental impairment in this population. This is an important study assessing the early childhood outcomes of late preterm infants, but a few issues need to be addressed before publication.
Methods
May improve clarity of methods to state all grades of IVH were included
It is not clear within the statistical methods what portions of the analysis the Mann-Whitney U test, Student t-test, and chi square test were used. In addition, the methods for the Youden index analysis presented in the results and the justification for its use in this type of study is not described in the manuscript.
The methods define significance as p<0.01; however results text defines the significance for Table 2 as p<0.05 and denotes male sex, hyperbilirubinemia, and severe hyperbilirubinemia as significant whose p-values would fall between these two thresholds. Please clarify as these variables are later utilized in subsequent analysis to determine clinical predictors of neurodevelopmental impairment.
Results
As this analysis involves follow-up data and the significant loss to follow up due to long-term data, it would strengthen the validity of the results to perform a comparison of the baseline characteristics of those with and without follow-up data available to demonstrate similarities and differences in those that presented for their neurodevelopmental assessment and those that did not. This information assists in demonstrating whether or not the studied population is representative of the late preterm population as a whole.
Section 3.2 appears to be inadvertently bold typed.
Discussion/Conclusion:
Given relatively small sample size and higher the loss to follow up rate (limitations that are both acknowledged in the manuscript), the wording of the abstract and manuscript conclusion stating that those with severe hyperbilirubinemia and/or IVH “must” be regarded as high risk for neurodevelopmental impairment is likely too strong of a statement to be supported by this data set. Recommend revising.
Author Response
The manuscript presents a retrospective analysis of the neurodevelopmental outcomes of late preterm infants. The study concludes that there is no significant difference between neurodevelopmental outcomes of 34 week infants as compared to those of 35 week infants when assessed at 18 months corrected age. Additionally, it concludes that severe hyperbilirubinemia and intraventricular hemorrhage are risk factors for neurodevelopmental impairment in this population. This is an important study assessing the early childhood outcomes of late preterm infants, but a few issues need to be addressed before publication.
Methods
May improve clarity of methods to state all grades of IVH were included
Per your suggestion, we have added relevant statements to improve the clarity of this in the Patients and Methods section (Pg 3, lines 116-117).
It is not clear within the statistical methods what portions of the analysis the Mann-Whitney U test, Student t-test, and chi square test were used. In addition, the methods for the Youden index analysis presented in the results and the justification for its use in this type of study is not described in the manuscript.
Accordingly, we have added relevant descriptions about the statistical methods used (Pg 3, lines 126-130, 133-135)
The methods define significance as p<0.01; however results text defines the significance for Table 2 as p<0.05 and denotes male sex, hyperbilirubinemia, and severe hyperbilirubinemia as significant whose p-values would fall between these two thresholds. Please clarify as these variables are later utilized in subsequent analysis to determine clinical predictors of neurodevelopmental impairment.
Thank you for your suggestions. Accordingly, we have added the clarification in the Patients and Methods section (Pg 3, line 124-126).
Results
As this analysis involves follow-up data and the significant loss to follow up due to long-term data, it would strengthen the validity of the results to perform a comparison of the baseline characteristics of those with and without follow-up data available to demonstrate similarities and differences in those that presented for their neurodevelopmental assessment and those that did not. This information assists in demonstrating whether or not the studied population is representative of the late preterm population as a whole.
Thank you for the advice. Based on your comments, we have added supplementary tables (S1, S2) to compare the baseline characteristics of those with and without follow-up data (Pg 4, lines 145-147). In addition, we confirmed that there were no significant differences between the two groups (both LPI-34 and LPI-35).
Section 3.2 appears to be inadvertently bold typed.
We apologize for the typographical errors; we have corrected this accordingly.
Discussion/Conclusion:
Given relatively small sample size and higher the loss to follow up rate (limitations that are both acknowledged in the manuscript), the wording of the abstract and manuscript conclusion stating that those with severe hyperbilirubinemia and/or IVH “must” be regarded as high risk for neurodevelopmental impairment is likely too strong of a statement to be supported by this data set. Recommend revising.
We apologize for the oversight, and we have revised this in the Conclusion to “might” (Pg 9, line 282).
Reviewer 2 Report
Nukasone et al. describes the neurodevelopmental impairments of infants born at 34 weeks compared with those at 35 weeks. They found no significant difference, which is not entirely surprising as the gestational difference between these groups is minimal. The following comments and suggestions will improve the manuscript:
- Methods: what gestational age is included at 34 or 35 weeks- is it 34+0-34+6 vs 35+0-35+6? This needs to be clarified and stated clearly
- Introduction: Ref 9; there is a suggestion of more immature renal function at 34 vs 35 weeks, however, this is not significant and if you review the paper there is a huge SD for levels. I am not aware that this marker has been validated as a marker of immaturity in preterm infants. Furthermore I am not sure how immature renal function relates to neurodevelopmental impairment?
- Introduction: you state that "As mentioned above, several reports 67 have recommended that LPI-34 be treated separately from LPI-35 or older." - however only one reference (ref 9) is provided, please provide more references to back up this statement.
- Introduction: you state "Therefore, we recalled the concept that 60 routine evaluation of the psychomotor development of uncomplicated LPIs could be limited to only 61 those at high risk for NDI, such as by limiting it to infants born at 34 GW.", where has this concept been recalled from? It is not something your unit does as you have collected this data from 35 weekers also. Please provide a reference here.
- Methods: You could consider expanding your 35 week group to include babies born up to 36+6? This may be more clinical relevant?
Author Response
Nukasone et al. describes the neurodevelopmental impairments of infants born at 34 weeks compared with those at 35 weeks. They found no significant difference, which is not entirely surprising as the gestational difference between these groups is minimal. The following comments and suggestions will improve the manuscript:
- Methods: what gestational age is included at 34 or 35 weeks- is it 34+0-34+6 vs 35+0-35+6? This needs to be clarified and stated clearly
We apologize for the unclear presentation of our manuscript. Accordingly, we have revised the Patients and methods section to improve its clarity (Pg 2, line 92-93)
- Introduction: Ref 9; there is a suggestion of more immature renal function at 34 vs 35 weeks, however, this is not significant and if you review the paper there is a huge SD for levels. I am not aware that this marker has been validated as a marker of immaturity in preterm infants. Furthermore, I am not sure how immature renal function relates to neurodevelopmental impairment?
We agree with the reviewer’s comment. Accordingly, we have deleted and revised the Introduction in Pg 2, line 70-71.
- Introduction: you state that "As mentioned above, several reports have recommended that LPI-34 be treated separately from LPI-35 or older." - however only one reference (ref 9) is provided, please provide more references to back up this statement.
Accordingly, we have added references 10-12 and revised the Introduction in Pg 2. Line 71-76, 78-79).
- Introduction: you state "Therefore, we recalled the concept that routine evaluation of the psychomotor development of uncomplicated LPIs could be limited to only those at high risk for NDI, such as by limiting it to infants born at 34 GW.", where has this concept been recalled from? It is not something your unit does as you have collected this data from 35 weekers also. Please provide a reference here.
We have recalled this concept from references 9-12. Accordingly, we have moved this statement to the last part of the third paragraph of the Introduction.
- Methods: You could consider expanding your 35 week group to include babies born up to 36+6? This may be more clinically relevant?
Thank you for your comment. Because we have routinely evaluated neurodevelopment of LPI born less than 36 weeks, we could not include these infants. Accordingly, we have revised the limitations (Pg 8, lines 258-259).
Reviewer 3 Report
Nakasone et al. presents an interesting manuscript in the obstetric field. The manuscript is well written with some grammatical errors, but it is necessary for the authors to make important changes:
-The abstract has to be rewritten. Authors should better describe their results.
-The introduction must include more current references. The spirit of the study must be justified with a more clinical perspective.
-The Material and Methods section should be named as Patients and Methods. The inclusion and exclusion criteria should be underlined. Authors must name the ethical references of their study.
-Better describe the statistical methodology used. This point is very incomplete and does not allow the reader to be certain of the conclusions.
-The sample size is very limited in a population as large as the Japanese. This is a critical point and the authors should name it as an important limitation of their study and discuss it.
-The authors must include the data of the placental characteristics of these patients and make a correlation. Placental alterations have been described in this field. This should be included in the discussion. The importance of vascular diseases should be included and described in the patients in the cohort.
-Figure 1 should have better quality. -The table feet must be described in more detail.
-The discussion must consider the most relevant aspects of fetal physiology. The placenta and its physiology must be taken into account. In this sense, vascular diseases have a lot to do.
-The authors should rewrite their conclusions to give it a more realistic perspective in daily obstetric practice.
-Authors should check their English grammar.
Author Response
Nakasone et al. presents an interesting manuscript in the obstetric field. The manuscript is well written with some grammatical errors, but it is necessary for the authors to make important changes:
-The abstract has to be rewritten. Authors should better describe their results.
Accordingly, we have revised the Abstract.
-The introduction must include more current references. The spirit of the study must be justified with a more clinical perspective.
Accordingly, we have added references 10-12 and have revised the Introduction in Pg. 2, lines 71-76.
-The Material and Methods section should be named as Patients and Methods. The inclusion and exclusion criteria should be underlined. Authors must name the ethical references of their study.
We have revised the relevant sections according to the reviewers’ suggestion (Pg 2, line 87 and pages 2-3, lines 91 to 96).
-Better describe the statistical methodology used. This point is very incomplete and does not allow the reader to be certain of the conclusions.
We apologize for the unclear presentation of our manuscript. Accordingly, we have revised the Statistical method (Pg 3, lines 124 to 141).
-The sample size is very limited in a population as large as the Japanese. This is a critical point and the authors should name it as an important limitation of their study and discuss it.
We agree with the reviewer’s comment. We have revised the limitation according to the suggestion. In addition, we added details on the sample size calculation to the method section.
-The authors must include the data of the placental characteristics of these patients and make a correlation. Placental alterations have been described in this field. This should be included in the discussion. The importance of vascular diseases should be included and described in the patients in the cohort.
Accordingly, we added the data of placental characteristics to Tables 1 and 2. In addition, we added the discussion regarding the importance of placental pathology to the limitations (Pg 9, line 266-273).
-Figure 1 should have better quality. -The table feet must be described in more detail.
Accordingly, we have revised Fig 1 to improve its quality.
-The discussion must consider the most relevant aspects of fetal physiology. The placenta and its physiology must be taken into account. In this sense, vascular diseases have a lot to do.
Accordingly, we have revised the Discussion in Pg 9, line 266-273).
-The authors should rewrite their conclusions to give it a more realistic perspective in daily obstetric practice.
Accordingly, we have revised our conclusions to address these concerns.
-Authors should check their English grammar.
We submitted our manuscript again for proofreading; it has been edited for language and grammar.

Round 2
Reviewer 2 Report
Thank you for your corrections and clarifying the methodology.
Author Response
Thank you for your insightful comment and suggestions. Our paper has been improved substantially. We are really appreciated.
Reviewer 3 Report
The authors have responded to all proposed comments.Authors must modify their manuscript more deeply, authors must make a single paragraph and expand the information.This point should be rewritten to attract the reader's attention.Authors should check for some spelling mistakes.
Author Response
The authors have responded to all proposed comments. Authors must modify their manuscript more deeply, authors must make a single paragraph and expand the information. This point should be rewritten to attract the reader's attention. Authors should check for some spelling mistakes.
Response 1:
Thank you for your valuable comment. In line with your suggestion, we have added a paragraph in the Discussion section (page 9 lines 269-277). In addition, we have got our manuscript checked and edited to correct the grammatical and typographical errors.